# Atmospheric electricity observations by Reinhold Reiter around Garmisch-Partenkirchen

R. Giles Harrison[a] and Kristian Schlegel[b]

[a]Department of Meteorology, Earley Gate, University of Reading, Reading. RG6 6ET UK

[b]Copernicus Gesellschaft, 3403 Göttingen, Germany

Correspondence to: Giles Harrison (r.g.harrison@reading.ac.uk)

submitted to *History of Geo and Space Sciences* (Atmospheric Electrical Observatories Special Issue)

**Abstract** Atmospheric electricity measurements were made at several sites close to Garmisch-Partenkirchen during four decades from 1950 to 1990 by Dr Reinhold Reiter, together with other environmental measurements. The quantities determined include the atmospheric potential gradient, the vertical current and the ion concentrations, and observations made at the Mount Wank site (1780 m, 47° 30' N, 11° 09' E) from 1st August 1972 to 31st December 1983 are available in digital form.

Keywords: Potential Gradient, conduction current; global circuit;

## 1.     Introduction

Motivated by his interest in the influence of atmospheric electric processes on humans, Reinhold Reiter (1920-1998) started atmospheric electricity measurements in the early 1950s. Past measurements of atmospheric electricity are increasingly studied internationally (Aplin, 2020), because of widening interest in the global atmospheric electric circuit and its relevance to climate (e.g. Nicoll et al., 2019). Data obtained in clean air conditions are of particular importance, such as from mountain sites. The atmospheric electrical quantities obtained by Reiter within a sustained campaign of environmental measurements frequently fulfilled the clean air requirements.

Reiter began with various measuring sites in Munich and southern Bavaria, probably to allow
intercomparisons. Later, he concentrated on measurements undertaken at Garmisch-
Partenkirchen, on the nearby Wank and Zugspitze mountains and onboard an instrumented
passenger cable car moving regularly between the Eibsee and the Zugspitze summit. (The
locations of these sites are shown in Fig. 1).

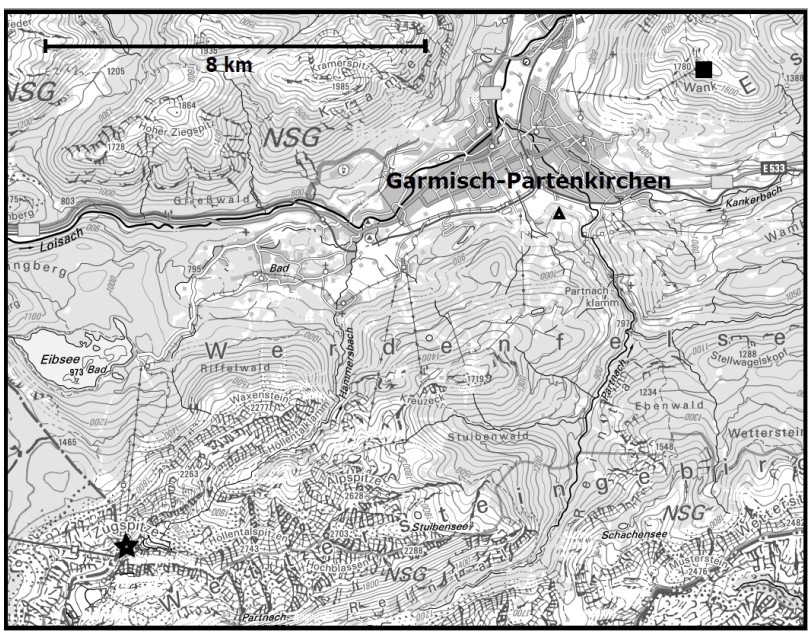

Figure 1: Area around Garmisch-Partenkirchen (southern Bavaria, Germany) with scale and
the observational sites marked: Wank (square, upper right), Central Institute (triangle) and
Zugspitze (star, lower left). The cable car runs almost directly north from the Zuspitze
summit to the right hand shore of lake Eibsee (map adapted from Digitale Topographische
Karte 1 : 100.000 (c) Bayerische Vermessungsverwaltung 2022, thanks to Martin Fasbender).

To undertake this, Reiter founded a privately-funded research institute, the *Physikalisch-*
*bioklimatische Forschungsstelle in Garmisch-Partenkirchen* which was incorporated as the
*Fraunhofer-Institut für Atmosphärische Umweltforschung* (IFU) in the Fraunhofer Society in
1962. He led this institute as its director until his retirement in 1985. In 2002 this institute
became part of the Institut für Meteorologie und Klimaforschung Atmosphärische
Umweltforschung (IMK-IFU), and Campus Alpin of the Karlsruher Institut für Technologie
(KIT).

Reinhold Reiter passed away on 24 September 1998 and a detailed memorial article was
published by Weihe (1999). It is understood that some possessions were bequeathed to Ettal
Abbey, a Benedictine monastery in Bavaria.

## 2.     Measurement locations

Reiter's principal scientific motivations were to investigate biometeorological responses to atmospheric variables such as the concentrations of small ions, and to study short-term solar-terrestrial influences on the global circuit. This may be reflected in the choice of mountain sites for the measurements, which brought the possibility of low pollution conditions and least local disturbances.

The Garmisch-Partenkirchen measurements were obtained at permanent sites on the Zugspitze (2964 m altitude) and Wank (1780 m) mountains, and at an additional site known as the Central Research Institute, on the valley floor (740 m). A novel feature was the use of the cable car connecting the Zugspitze and a ground station close to lake Eibsee, instrumented to carry sensors in a regular path, sometimes passing repeatedly through fog and cloud layers. Vertical profiles of ozone were obtained using this approach (Reiter, 1991).

## 3.     Apparatus

Customised instruments and systems were devised for the atmospheric electrical measurements. A primary quantity studied was the vertical potential gradient (PG). On Wank, as well as on the cable car, a radioactive collector probe was used, connected to a high impedance electrometer amplifier. The PG sensing probe was heated, and its physical construction refined during a long period of operation in mountain conditions, especially precipitation. The atmospheric conductivity was measured with an aspirated Gerdien condenser. A further measurement of the PG was made using an electrostatic field mill, and the air-earth current with a wire antenna. A special device was developed for measuring the space charge and, simultaneously, the natural radioactivity in the air. Beyond the usual fair weather measurements, the precipitation current density was obtained with an electric rain gauge. All these instruments and corresponding results are described in papers (Reiter, 1977a, b), and Reiter's textbook (Reiter, 1992).

## 4.     Data recovery

Some of the measurements from the Bavarian Alps have previously been made available on a CDROM, which was originally distributed through the collaborative network provided by the SPECIAL scientific community (Rycroft and Füllekrug, 2004). These data values were

retrieved from magnetic tapes in summer 2000, with the help of one of Reiter's collaborators.
They provide hourly values from the Wank site (1780m, 47°30'N, 11°09'E), and span 1st
August 1972 to 31st December 1983. The wide range of quantities recorded is summarised in
Table 1, with the atmospheric electricity quantities identified.
**Table 1. Quantities recorded on Mount Wank (1972-1983)**

| Description of measured quantity | Symbol used in dataset |
|---|---|
| **Meteorological and Environmental** | |
| air temperature | T |
| relative humidity | RF |
| water vapor partial pressure | E |
| specific humidity unit | SF |
| potential temperature | TH |
| equivalent potential temperature | THE |
| wind speed | WG |
| wind direction | WR |
| Sunshine duration | SD |
| Global solar irradiance | GS |
| Sky radiation | HS |
| UV intensity | UV |
| **Atmospheric Electrical** | |
| Electric field | F |
| Zero crossing of F | DU |
| Vertical current | I |
| Positive ion concentration | N+ |
| Negative ion concentration | N- |
| Total ion concentration | SN |
| Positive ion conductivity | L+ |
| Negative ion conductivity | L- |
| Total ion conductivity | SL |

| Number concentration of condensation nuclei | K1, K2, K3 |
| --- | --- |


### 5.    Discussion

The PG measurements obtained were over a sufficiently extended period to provide statistical
support for suspected solar effects on the lower atmosphere (Reiter, 1977b), which was a major
topic of research interest in the 1970s (e.g. Olson, 1971). Due to these effects emerging, it is
likely that the local influences are sufficiently small that global atmospheric electric circuit
variations can also be retrieved.

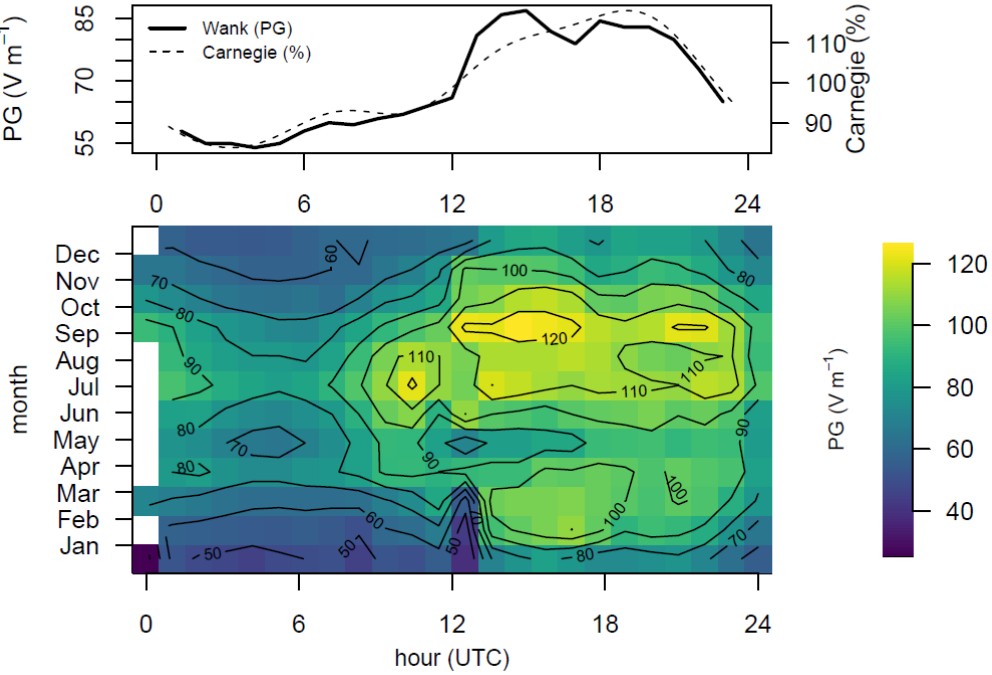


Figure 2. *Upper panel:* Median hourly Potential Gradient (PG) across all months of the year
from Mount Wank, with the relative variation from Cruise VII of the *Carnegie* overplotted.
*Lower panel:* Hourly median PG by month from Mount Wank, using values for 1976-1983.

Fig 2 provides a summary of the seasonal and diurnal variation in the PG at the site using data
from 1976 onwards, which is the longest period of consistent data following an unexplained
step change in the mean values. The upper panel of fig 2 shows the hourly variation across all
months, which is compared with the well-known global circuit "Carnegie curve" variation.
Although there are discrepancies in detail, perhaps arising from local meteorological factors or
uneven sampling, the (Pearson) correlation between hourly values of the Carnegie curve and
the Mount Wank PG is 0.96. The probability $p$ that this is due to chance is small ($p<0.001$),
using the method of Ebisuzaki (1991) which accounts for serial correlation. The lower panel
of fig 2 shows the diurnal variations by month, in which the Carnegie curve is evident more
strongly in the second half of the year. Some values around midnight UTC are absent.

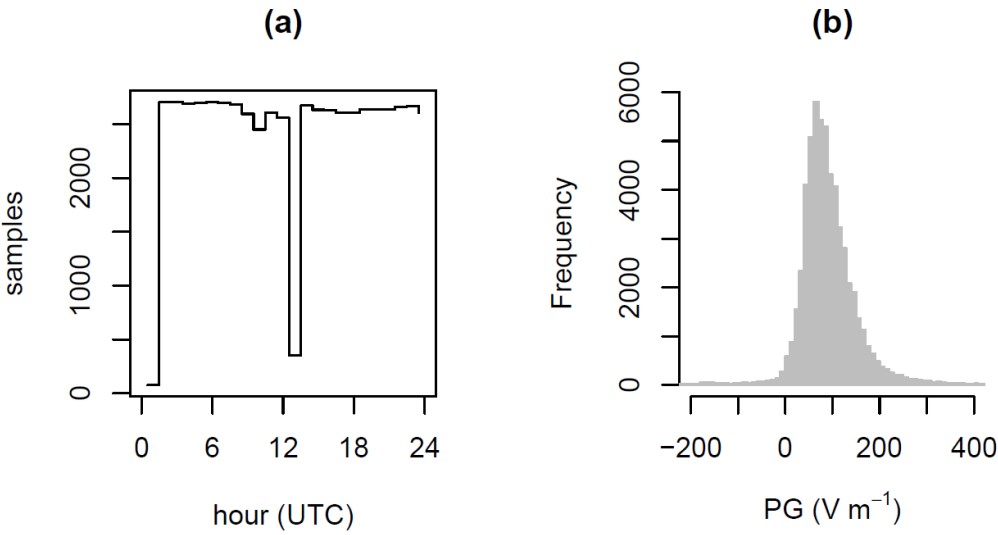

Figure 3. (a) Count of hourly samples of Mount Wank PG values 1976-1983 and (b)
distribution of all the hourly PG values obtained.
Fig 3 summarises the sampling and the distribution of values obtained. From fig 3a it can be
seen there are far fewer values for midnight and midday than for any of the other hours. It is
not clear why this is, but both midnight and midday occur first in each line of values in the data
files, so it might be a data processing artifact. A similar pattern of missing values is found for
some other measured quantities in the data files. Fig 3b presents the combined hourly PG data
as a histogram: the median is 84 Vm$^{-1}$, and interquartile range 58 Vm$^{-1}$ to 119 Vm$^{-1}$.
Fig 4 demonstrates the consistency evident between annual variations in PG measurements
from Mount Wank, and those made at Lerwick, Shetland (Harrison and Riddick, 2022), for
Decembers which have values available digitally. Some of the variations observed at Shetland
are thought to arise from the El Niño-Southern Oscillation (Harrison et al, 2022), in turn
modifying the global distribution of current-generating storms. Fig 4a shows the values as a
time series. Although there is a trend in the Mount Wank data (Harrison, 2004), fig 4b shows
the correlation between the two short series of values. This is consistent with the global circuit
providing the common variations occurring at both sites.

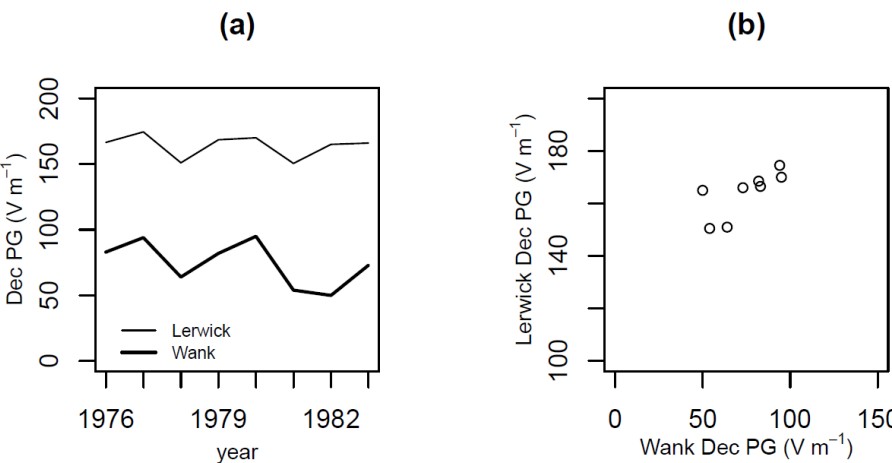


Figure 4. Annual December mean PG values for Mount Wank compared with those for
Lerwick, following Harrison (2004), as (a) time series and (b) a scatterplot. The correlation
coefficient $r$ in (b) is $r = 0.74$ ($p = 0.03$).

Combined with the Carnegie curve agreement of fig 2, fig 4 further supports the value of the
Mount Wank PG data for studying global circuit effects.
**6.     Conclusions**
Atmospheric electricity and other environmental measurements were made in the Bavarian
Alps over a long period, from which a series of hourly measurements for much of the 1970s is
available digitally. In the PG data from the Mount Wank site, the presence of global and solar-
terrestrial signals is apparent, which indicates the likely wider applicability of the
measurements. The endeavours at the Garmisch-Partenkirchen sites deserve to be more widely
known.

**Data availability**
The 1972-1983 Wank dataset is openly accessible through the University of Reading's
Research Data Archive, at https://doi.org/10.17864/1947.000445 . (The Lerwick December
data is available at https://doi.org/10.17864/1947.000409 ).


**Author Contributions**
The authors jointly drafted the manuscript.

**Competing interests**
Kristian Schlegel is an editorial board member of HGSS. There are no other competing
interests.

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
