# Peer review of "Atmospheric electricity observations by Reinhold Reiter 1 around Garmisch-Partenkirchen 2"

_History of Geo- and Space Sciences, 2023_

## Author Response (AR1)

**Atmospheric electricity observations by Reinhold Reiter around Garmisch-Partenkirchen**

R. Giles Harrison and Kristian Schlegel https://doi.org/10.5194/hgss-2023-4
* * *
We thank both reviewers for the careful consideration of our manuscript and the helpful points made. Our manuscript has been revised accordingly, and the principal changes made in response to the reviewers' comments are provided on the marked copy in red.

**Reviewer 1**

*Firstly, I would like to thank the Authors for submitting their valuable research. Making historical data available is beneficial for all the scientific community. The manuscript is concise and logically structured. It contains six short sections introducing the research topic, the data, the measurement sites, and apparatus; describing the recovered parameters; discussing some results, and ultimately drawing the conclusions. The manuscript is fully in line with the scope of the journal and makes an important scientific contribution, therefore, I recommend it for publication with listing some minor comments below which the Authors can consider implementing.*

We thank the reviewer for considering our manuscript carefully and for the positive points made.

*Specific comments:*

*1, Fig. 1.: It would make the map easier to understand if you could include a scale bar indicating the distances.*

We agree and we have revised fig 1 to include a scale.

*2, Fig. 2/a.: Please, include the unit (%?) of the relative Carnegie curve on a second y axis and specify the time zone on the x axis (Coordinated Universal Time?).*

Thank you. We have revised this figure more substantially, and we have specifically included the relative Carnegie variation as requested. The original Carnegie data was reported against times which were recorded in GMT, but using UTC, as suggested, is equivalent and widely understood.

*3, Fig. 2.: Correlations coefficients could be included both on panel (a) (between the Wank and Carnegie curves) and on panel (b) (between the Lerwick and Wank curves).*

For the original figure 2, the correlations $r$ and the probabilities $p$ of chance correlation (using the Ebisuzaki method, now cited) are (a) $r=0.96$ ($p=0.0002$) and for (b) $r=0.83$ ($p=0.03$). For the revised version we have added these values to the text.

*4, Section 5, Discussion: Could you please describe briefly why you used only December values?*

So far, only the December values have been keyed for Lerwick, and these values are readily available in the University of Reading data repository. We mention this aspect of the digital data availability in the revised manuscript.

*5, Including a histogram of the PG data as a separate figure could add more information to section 4. Data recovery (or including it as a subplot in Fig. 2.).*

We agree and we have chosen to provide this histogram as an additional figure to summarise the range of the values, which complements the time series information provided in the 2004 paper and the information included in the data repository submission.

**Reviewer 2**

*Harrison and Schlegel have made available meteorological and electrical parameters recorded at Mount Wank for the period 1 August 1972 to 31 December 1983. The article is well written, clear and worthy of publication. However, in the reviewer's opinion, it would be good if the authors could clarify a couple of points in the manuscript.*

Thank you for the thorough attention given to the paper and the positive comments made.

*1. Section 3: PG measurements were recorded with two different sensors (radioactive collector probe and electric field mill). It was not clear to this reviewer if the available measurements are made with these two sensors, or if it was first made with one sensor and then with the other sensor?........... Is this the explanation why there is a step change in the PG measurements after March 15, 1976?*

The origin of this step change is unfortunately not known, nor is the precise use of the different sensors. We now mention the existence of the step change in the revised text.

*2. Section 5: In the opinion of this reviewer the discussion section should be improved. My suggestion is to go further into the figures. For example:*

*Figure 2a: Why the Wank curve has a peak at ~14-15 UT (clearly different from the maximusm peak of the Carnegie curve). Maybe the method using the median is not the most appropriate?*

Local meteorological factors can influence the values at different times of day. We have considered the possibility that the time synchronisation is inaccurate, but the decreases at the end of the day in both curves are exactly coincident, suggesting that this is not the core explanation. Irregular sampling is another possibility, which we now mention in the revision. Use of the mean instead of the median (below), also does not resolve the question.

[Figure]

Figure 2. Examples of Potential Gradient (PG) data from the Wank site. (a) Hourly **mean** PG using December-only values, 1976-1983 (solid line), overplotted on (relative) Carnegie curve for November-December-January. (b) Annual December values for Lerwick (thin line) and Wank (thick line), from Harrison (2004).

Clearly, a rather more extensive investigation is needed, but our object here is primarily to ensure the data is known and available to other researchers rather than fully analyse it. Nevertheless, we have revised the figure to summarise the diurnal variation more completely across all the months, whilst also retaining the Carnegie curve for comparison.

*Figure 2b. What is the explanation of the 2 PG minimima values found at both sites in 1978 and 1981? The decline trend in PG values at Wank is greater than Lerwick site, is this due to an instrumental effect?*

Recent work using December PG data from Lerwick (Harrison et al, Environ Res Lett 17, 124048, 2022) has shown that, in the interval concerned, the variations yielding the PG variations arise from the El Niño-Southern Oscillation, through its effect on modifying the global distribution of current-generating storms. This work was not published when the draft manuscript was written, but it is now mentioned in the revised manuscript.

For the trend, there could be many sources, such as global circuit changes reduction in aerosol loading, or, as suggested, an instrumental effect. This was, to some extent, addressed in the 2004 paper, but inconclusively: it is mentioned in the revised text.

*Minor comment: Line 32: Reiter* - Thank you.